# Conservative Scoring Approach in Productivity Susceptibility Analysis Leads to an Overestimation of Vulnerability: A Study from the Hilsa Gillnet Bycatch Stocks of Bangladesh

**Hasan Faruque [1,2,*] and Hiroyuki Matsuda [1]**

1    Graduate School of Environment and Information Sciences, Yokohama National University,
     Yokohama 240-8501, Japan; matsuda-hiroyuki-vj@ynu.ac.jp
2    Department of Fisheries, Faculty of Biological Sciences, University of Dhaka, Dhaka 1000, Bangladesh
*    Correspondence: faruque-hasan-ny@ynu.jp or hasanfaruque28@du.ac.bd

**Abstract:** Despite different approaches used to assign the risk scores for missing information in productivity susceptibility analysis (PSA)—a widely used semi-quantitative risk assessment tool for target and non-target fisheries stocks—for the selected attributes of a given species, no formal comparison has been made between scoring approaches in terms of how well they can predict species vulnerability. The present study evaluated the PSA findings of 21 bycatch stocks of the Hilsa (*Tenualosa ilisha*) gillnet fishery of Bangladesh using two different scoring approaches (the conservative scoring approach, CSA; and the alternative scoring approach, ASA) to determine the most reliable approach to minimize false estimates of species vulnerability. Our analysis revealed that the *V* scores increased by $0.0-0.20$ with a mean value of 0.09 for 21 selected bycatches when CSA was applied. The inconsistency between the vulnerability (*V*)-score-suggested fishing status ($V \leq 1.8$ = underfishing, $V > 1.8$ = overfishing) and the fishing status defined by exploitation rate ($E > 0.5$ = overfishing, $E < 0.5$ = underfishing) were 38.1% and 19.0% under CSA and ASA, respectively. Likewise, the consistency between the *V*-score-suggested fishing status and fishers' perceived catch trends was found to be higher when using ASA than when using CSA. Our analysis suggests that CSA could overestimate species vulnerability. Therefore, ASA is more reliable than CSA in PSA, which may increase the confidence of fisheries stakeholders in PSA.

**Keywords:** *Tenualosa ilisha*; Indian shad; gillnet fishery; data-limited fishery; bycatch stock; risk assessment; precautionary approach; life-history parameters

## 1. Introduction

The sustainable management of fisheries resources is a challenging issue for fisheries managers across the world [1]. Fisheries management benefits from accurate stock status estimates to apply harvest control rules and meet management objectives [2,3]. The stock status compared to different biological reference points (e.g., maximum sustainable yield) can be adequately made by conventional quantitative stock assessment method, particularly in data- and capacity-rich settings [4,5].

Generally, large-scale fisheries target species with high commercial value. These species are subject to more detailed analyses of their life-history traits, productivity, etc., and are recognized as data-rich stocks. In contrast, the majority of small-scale fisheries, which account for half of the global fishery catches, are treated as data-limited fisheries [6,7]. These small-scale fisheries lack the biological and catch data, resources, and expertise required to estimate stock status using conventional quantitative stock assessment techniques [6]. Therefore, the actual statuses of most global fish stocks from small-scale fisheries remain unknown [8]. Such fisheries remain unmanaged or managed with insufficient scientific guidance, leading to suboptimal catch rates and adverse social and economic consequences for those who depend on fishing [9]. These cases are particularly evident in tropical and

subtropical regions where multi-species and multi-gear fisheries exist, and diverse groups of species are often discarded or retained as bycatches with low commercial value [10].

Fishing activities, by definition, have a direct effect on the abundance of targeted fish stocks and populations and may also have a negative effect on the status of bycatch stocks [11]. While bycatches are recognized as an important biological component of the ecosystem, bycatch stock status is insufficiently assessed using traditional quantitative stock assessment methods [12] due to a lack of information (e.g., time series catch and effort data, life history data, etc.) [13,14]. Following the increased need to address fishing's impacts on the whole range of exploited stocks, including bycatch species, fishery scientists have sought to develop comprehensive methods to assess the potential risk of various fishing types (gillnet fishing, seine net fishing, longline fishing, etc.) in data- and capacity-constrained situations where quantitative assessment is not feasible due to data scarcity [15,16].

Risk or vulnerability assessment typically follows a semi-quantitative approach for data-limited stocks [3]. The semi-quantitative methods designed for evaluating fisheries' impacts on target or bycatch stocks [17,18], extinction risk [19,20], and impacts on ecosystem sustainability [21,22] typically facilitate the inclusion of both qualitative and quantitative information and a wide range of variables. One of the most widely recognized and used semi-quantitative assessment tools is called Productivity Susceptibility Analysis (hereafter referred to as PSA) [23,24]. The PSA is currently being used and recommended by several fisheries management agencies, including the AFMA (Australian Fisheries Management Authority), ICCAT (The International Commission for the Conservation of Atlantic Tunas), IOTC (Indian Ocean Tuna Commission), MSC (Marine Stewardship Council), NMFS (National Marine Fisheries Service, USA), and WCPFC (Western and Central Pacific Fisheries Commission) [14,23–25]. Thousands of stocks and populations across the world, including target and bycatch fish stocks, sea birds, sea turtles, squids, octopus, and marine mammals, have already been assessed by PSA [26,27].

The most general feature of PSA is that it compares the inherent recovery potential of species once depleted (i.e., productivity attributes) with the attributes of susceptibility (i.e., the impact of the fishery on fish stock) to fishing activities in elucidating overall vulnerability [18,24]. Since its first use in 2001 for evaluating the risk of an Australian Prawn fishery in terms of bycatch stocks, different modifications and improvements have been made to the PSA tool. These include increases in the number of attributes rated, the development of additive methods for calculating the weighted average score for productivity and susceptibility attributes, the inclusion of a five-tier data quality index, and the ability to test a range of alternative approaches for missing data [23]. Different scoring approaches, moreover, have been used by scientists to treat the missing data in PSA. One approach is to assign a score representing high risk when the data for a particular attribute is missing. This approach is known as the "precautionary or conservative scoring approach" in PSA [24]. In contrast, some authors have removed the missing attributes from PSA, and finally, PSA findings were interpreted using data-quality ratings [3,23]. Most recently, different empirical equations have been used to derive data from correlated life-history attributes when scoring the missing data for a particular attribute [14,28]. For instance, the von Bertalanffy growth coefficient (*k*; how rapidly a fish reaches its maximum size) tends to be strongly related to fish's maximum age. Stocks with a long lifespan and low productivity tend to have a high *k*-value [29]. In this way, it is possible to obtain the values for the growth coefficient of fish (if data on the growth coefficient is missing) by using an empirical relationship between the growth coefficient and the maximum age of the fish. While different approaches have been used to assign the scores for missing data for the attribute(s), to the best of our knowledge, no formal comparison has been made between the different scoring approaches for evaluating how well these approaches predict species vulnerability to fishing activities by judging the PSA outcomes through other analytical assessments (e.g., the exploitation rate, which indicates the overfishing or underfishing status of stocks, catch trends, etc.)

Under this background, the present study compared the results of PSA for the bycatch stocks of the Hilsa (*Tenualosa ilisha*) gillnet fishery of Bangladesh using two different scoring approaches to determine a more reliable and advisable approach that can reduce false estimates of species vulnerability. Two scoring approaches used in the PSA analysis were designated as conservative scoring approaches (CSAs), which assign the highest risk score based on missing information, and alternative scoring approaches (ASAs), which include expert opinions and/or the usage of an empirical relationship equation to derive missing data when the values of the correlated parameters are known, particularly for productivity attributes.

## 2. Materials and Methods

### 2.1. Selection of Bycatch Species from the Hilsa Gillnet Fishery in Bangladesh

Hilsa shad (*Tenualosa ilisha*), which constitutes the important fishery in the Bay of Bengal and Persian Gulf region, is the single most dominant species in Bangladeshi waters [30]. This transboundary species largely migrates from seawater to the estuarine and riverine ecosystem during its spawning time, and it is largely captured mostly by gillnets [31]. Gillnet fishing accounts for over 95% of the Hilsa catch in Bangladesh, which supports over 2.5 million peoples' livelihoods [32].

Hilsa fishers mainly focus on Hilsa as their target species. However, many other fishes are being captured by their gillnets due to the less selective nature of the gillnet itself and the multi-species characteristics of Bangladeshi fisheries. Faruque and Matsuda [33] have recently identified and reported 129 bycatch species from Hilsa gillnet fishing in Bangladesh. This study considered 21 bycatch species from the Hilsa gillnet fishery for their vulnerability analysis with PSA in two different scoring approaches. The species we have selected for PSA are given in Table 1.

**Table 1.** List of bycatch species from the Hilsa gillnet fishery of Bangladesh for vulnerability assessment with PSA.

| Scientific Name | FAO Species Code | Common Name | Family | Order | Environment Preference |
|---|---|---|---|---|---|
| *Clupisoma garua* | LUG | River catfish | Ailiidae | Siluriformes | Freshwater, brackish |
| *Coilia ramcarati* | ZZU | Ramcarat grenadier anchovy | Engraulidae | Clupeiformes | Marine, brackish |
| *Harpadon nehereus* | BUC | Bombay-duck | Synodontidae | Aulopiformes | Marine, brackish |
| *Ilisha filigera* | PIF | Coromandel ilisha | Pristigasteridae | Clupeiformes | Marine, freshwater, brackish |
| *Lates calcarifer* | GIP | Giant perch | Centropomidae | Carangiformes | Marine, freshwater, brackish |
| *Lepturacanthus savala* | SVH | Savalani hairtail | Trichiuridae | Scombriformes | Marine, brackish |
| *Megalaspis cordyla* | HAS | Torpedo scad | Carangidae | Carangiformes | Marine, brackish |
| *Mystus gulio* | BMG | Long whiskers catfish | Bagridae | Siluriformes | Freshwater, brackish |
| *Nemipterus japonicus* | NNJ | Japanese threadfin bream | Nemipteridae | Perciformes | Marine |
| *Netuma thalassinus* | AUX | Giant catfish | Ariidae | Siluriformes | Marine, freshwater, brackish |
| *Otolithoides pama* | OTD | Pama croaker | Sciaenidae | Perciformes | Marine, freshwater, brackish |
| *Pampus argenteus* | SIP | Silver pomfret | Stromatidae | Scombriformes | Marine |
| *Pampus chinensis* | CPO | Chinese silver pomfret | Stromatidae | Scombriformes | Marine, brackish |
| *Parastromateus niger* | POB | Black pomfret | Carangidae | Carangiformes | Marine, brackish |
| *Pennahia argentata* | CRV | Silver croaker | Sciaenidae | Perciformes | Marine |
| *Polynemus paradiseus* | ONU | Paradise threadfin | Polynemidae | Carangiformes | Marine, freshwater, brackish |
| *Pomadasys argenteus* | GRL | Silver grunt | Haemulidae | Perciformes | Marine, freshwater, brackish |
| *Rastrelliger kanagurta* | RAG | Indian mackerel | Scombridae | Scombriformes | Marine |
| *Rhinomugil corsula* | RIC | Corsula mullet | Mugilidae | Mugiliformes | Freshwater, brackish |
| *Scoliodon laticaudus* | SLA | Spadenose shark | Carcharhinidae | Carcharhiniformes | Marine, brackish |
| *Scomberomorus guttatus* | GUT | Indo-Pacific king mackerel | Scombridae | Scombriformes | Marine, brackish |

The selected bycatch species belong to eight different orders in 17 families and most commonly inhabit marine and brackish water ecosystems, with some from freshwater habitats. We selected these 21 bycatch stocks for vulnerability evaluation because the exploitation status of these species from Bangladeshi waters was previously assessed from length-based data using a quantitative stock assessment tool (FAO-ICLARM stock assessment tools) (Table S1). We compared this formal assessment outcome (i.e., exploitation rate, *E*) with our PSA assessment outcome to determine the consistency or inconsistency rate between two outcomes under two different scoring approaches, as described in Section 2.6.

### 2.2. Selection of Productivity (P) and Susceptibility (S) Attributes for PSA

Flexibility in selecting the number of attributes makes the PSA more compatible than other semi-quantitative vulnerability assessment tools [23]. The selection of attributes for productivity or susceptibility scoring mainly depends on the availability of the data and its ability to represent vulnerability. However, a greater selection of attributes can help ensure that a sufficient number of attributes are rated [3]. We considered 12 productivity attributes (Table 2) and 10 susceptibility attributes (Table 3) in our study.

**Table 2.** Productivity attributes and their scoring criteria were used to determine the productivity of the selected bycatch stocks from the Hilsa gillnet fishery in Bangladesh (adopted from Faruque and Matsuda, 2020).

| Productivity Attributes | Low Risk (3) | Moderate Risk (2) | High Risk (1) |
|---|---|---|---|
| Maximum age ($t_{max}$, year) | <4 | 4–8 | >8 |
| Maximum size ($L_{max}$, cm) | <38 | 38–85 | >85 |
| Von Bertalanffy growth coefficient ($k$, yr$^{-1}$) | >0.78 | 0.33–0.78 | <0.33 |
| Estimated natural mortality ($M$, yr$^{-1}$) | >1.21 | 0.74–1.21 | <0.74 |
| Measured fecundity (MF) | >64136 | 10663–64136 | <10,663 |
| Breeding strategy (BS) | Release eggs into the water column | Lay eggs in a nest and guard those eggs until hatching | Internal fertilization (/Livebearer) mouth brooding or other strategies that involve full parental care |
| Age at first maturity ($t_{mat}$, years) | <1.0 | 1–2 | >2 |
| Mean trophic level (MTL) | <3.50 | 3.50–3.90 | >3.90 |
| Size at first maturity ($L_{mat}$, cm) | <19 | 19–38 | >38 |
| Breeding cycle (female) | Annual cycle with protracted breeding season | Annual cycle with a seasonal peak | Bi/Triennial |
| $t_{mat}/t_{max}$ | <0.25 | 0.25–0.30 | >0.30 |
| $L_{mat}/L_{max}$ | <0.52 | 0.52–0.59 | >0.59 |

The productivity of a species or population is heavily influenced by its intrinsic characteristics [24]. Among the 12 selected productivity attributes, the first eight attributes (e.g., maximum age, growth coefficient, and natural mortality) were taken from Patrick et al. [3]. These eight attributes are commonly used in PSA. Each of the selected attributes has an influence on species productivity. The remaining four productivity attributes—size at maturity [24], breeding cycle [34], maturity–size ratio, and maturity–age ratio [35]—were obtained from other works because of their strong correlation with the productivity of the stocks. Some attributes (maximum age, maximum size, and age and size of fish at maturity) are negatively correlated with species productivity, which means that species that attain a larger size, longer lifespan, and slower growth rate are less productive. Conversely, some attributes are positively correlated with population productivity (e.g., species with greater natural mortality tend to spawn more eggs to replenish the loss) [36]. Likewise, among the 10 susceptibility attributes, the first eight attributes, which are commonly used in PSA (e.g., vertical overlap, seasonal migrations, management strategy, etc.), were chosen from Patrick et al. [23]. The market value of fish (USD/kg) and the market demand for fish were taken from Faruque and Matsuda [33].

**Table 3.** A set of attributes and their scoring criteria were used to determine the susceptibility of the selected bycatch stocks from the Hilsa gillnet fishery in Bangladesh (adopted from Faruque and Matsuda, 2020).

| Susceptibility Attributes | High Risk (3) | Moderate Risk (2) | Low Risk (1) |
|---|---|---|---|
| Areal overlap | >50% of the stock occurs in the area fished | Between 25% and 50% of the stock occurs in the area fished | <25% of stock occurs in the area fished |
| Vertical overlap | >50% of the stock occurs in the depths fished | Between 25% and 50% of the stock occurs in the depths fished | <25% of stock occurs in the depths fished |
| Seasonal migrations | Seasonal migrations increase overlap with the fishery | Seasonal migrations do not substantially affect the overlap with the fishery | Seasonal migrations decrease overlap with the fishery |
| Schooling, aggregation, and other behavioral responses | Behavioral responses increase the catchability of the gear | Behavioral responses do not substantially affect the catchability of the gear | Behavioral responses decrease the catchability of the gear |
| Morphological characteristics affecting capture | Species shows high selectivity to the fishing gear (e.g., torpedo-shaped or bilaterally flattened with deeper girth fishes) | Species shows moderate selectivity to the fishing gear (e.g., elongated body shaped fishes) | Species shows low selectivity to the fishing gear (e.g., flatfishes) |
| Management strategy | Stocks do not have input and/or output control measures, and target and bycatch species are not monitored | Stocks have input and/or output control measures, and measures in place to conserve the stocks occasionally monitored and enforced | Stocks have input and/or output control measures, and measures in place to conserve the stocks regularly monitored and enforced by balancing carrots and sticks |
| Survival after capture and release | Probability of survival <33% | Between 33% and 67% probability of survival | Probability of survival >67% |
| Market value of fish (USD / kg) | >3.5 | 1.5–3.5 | <1.5 |
| Market demand for fish | High | Moderate | Low |
| Fishing rate relative to natural mortality | >1 | 0.5–1.0 | <0.5 |

Some biological parameters (e.g., maximum age and age at first maturity, maximum size, and size at first maturity) are highly correlated with each other. Therefore, the possibility of autocorrelation among the selected attributes cannot be ignored [28]. The weighting for the biological parameters of the fish defined primarily by the productivity attributes can be increased implicitly if double counting occurs. It was previously suggested to exclude the attributes where correlation exists, and the value of the correlation coefficient is as high as 0.90 [24]. Our correlation matrix among the attributes showed no set of attributes for which the correlation coefficient was greater than 0.90, except for the attributed maximum size and size at first maturity. However, the exclusion of either of these two attributes did not significantly change the vulnerability score or category. Therefore, we left both attributes in our analysis.

### 2.3. Data Collection for Attribute Scoring

Data on the productivity attributes (e.g., $L_{max}$, $k$, $M$, MF, and BS) were mostly collected from published journal articles, grey literature, and books (see Table S3). We prioritized species-specific data collection from Bangladeshi water areas wherever possible. We also considered the attribute information, especially for information on the MF and BC attributes of some species, for members of the same genus in Bangladesh or the Indian subcontinent, or globally as appropriate, when species-specific data were unavailable [37]. In cases where information was unavailable for some particular attributes, such as $t_{max}$, $t_{mat}$ and $L_{mat}$, of a given species, we considered the empirical relationships [29,38] between the attributes to calculate the missing attribute values from the values of known attributes of same species based on the assumption that some bi-

ological parameters of fish are highly correlated [39–41]. Lin et al. [28] and Faruque and Matsuda [33] used similar types of approaches in their assessments. For example, the equation of $t_{max} = 3/k$ ($t_{max}$ = maximum age; $k$ = the von Bertalanffy growth coefficient) was used to estimate $t_{max}$ from the available data on $k$. We also considered the following equations to calculate the age at first maturity ($t_{mat}$) and length at first maturity ($L_{mat}$): $t_{mat} = $ -$\log_e$ $(1$-$L_{mat}/L_\infty)/k$ ($L_\infty$ = asymptotic maximum length) and $L_{mat} = L_\infty 10^{(0.8979-0.0782T)}$ (T = water temperature), respectively. Information on the "mean trophic levels" of all assessed bycatch stocks was borrowed entirely from the online open-access library FishBase [42].

The information on the susceptibility attributes was also collected from published articles, reports, and books (Table S4). In addition, data on the market demand and selling prices of bycatch species, gillnet selectivity to bycatch species, fishing areas and times, gillnet-deployed water depth, gillnet dimensions and mesh sizes, the tendency of fishers to release non-target species back into the water, fishery rules and regulations and their effectiveness, and the fishery's degree of compliance with fishery laws were mainly collected directly from field observations, in-person interviews, and focus group discussions with experienced and knowledgeable Hilsa fishers (i.e., those with at least 10 years of Hilsa fishing experience). The bycatch species data considered for the PSA in our study were reported from the inland and marine Hilsa habitats of Bangladesh. In total, 50 Hilsa gillnet fishers from an inland habitat adjacent to the Hilsa hotspot rivers (e.g., Meghna, Padam, Tetulia, Andharmanik, and Galachipa) and 50 Hilsa gillnet fishers from a marine habitat (e.g., Bay of Bengal) were selected using the judgmental sampling technique [43], also known as purposive sampling, for face-to-face interviews and focus group discussions, mainly to gather information on the Hilsa fishery of Bangladesh. Specific survey points of the inland and marine habitats are provided in Table S2. The information gathered on the Hilsa fishery and its bycatch stocks from interviews and direct observations was used to score some of the susceptibility attributes (vertical overlap, management strategy, bycatch species survival after release, management strategies, etc.).

The yearly catch data for the selected bycatch species were not available, except for data on *Harpodon nehereous* [32] from Bangladeshi waters. Therefore, to obtain qualitative information on the bycatch species' catch trends, we asked the Hilsa fishers to score the bycatch species on a scale of 1–3, with "1", "2", and "3" denoting decreasing, stable, and increasing trends, respectively (Table S1). This species catch trend information was used to compare the vulnerability scores, as described in Section 2.6 (comparison of the species *V* score with *E* and the catch trend).

### 2.4. Conservative Scoring and Alternative Scoring Approaches

Typically, in PSA, all the productivity and susceptibility attributes are ranked on an ordinal scale. In this ordinal scale (i.e., a 1–3 scale), the scores "1", "2", and "3" represent the "low", "moderate", and "high" productivity and susceptibility of stocks. Bycatch stocks with low *P* and high *S* scores represent high vulnerability due to Hilsa gillnet fishing, whereas bycatch stocks with high *P* and low *S* scores indicate low vulnerability. In the conservative scoring approach, we assigned the lowest score to *P* and the highest score to *S* (i.e., the highest risk) when data were missing, as done in Hobday et al. [24].

Alternatively, to collect missing information, we incorporated expert opinions (e.g., local fishery officials through key informant interviews) and used the empirical relationships (described in Section 2.3) between the productivity attributes (see Tables S3 and S4). The use of this approach for treating missing data while scoring the attributes was called the "alternative scoring approach" in our PSA. The scoring thresholds for quantitative data ($t_{max}$, *M*, etc.) and scoring criteria for qualitative data (management strategy, market demand for fish, etc.) were retained from Faruque and Matsuda [33] (Tables 2 and 3). All the attributes were equally weighted with default values of 2, as in Patrick et al. [24]. We referred to Faruque and Matsuda [33] for further details on how to determine scoring thresholds and set criteria for the bycatch stocks of the Hilsa gillnet fishery in Bangladesh.

The data used for scoring each of the productivity and susceptibility attributes and the assigned scores with the data references are provided in the Supplementary Materials (Tables S3 and S4).

### 2.5. Determination of Bycatch Stocks' Vulnerability (V)

Vulnerability (*V*) refers to the degree to which a species' biological capacity to regenerate is outstripped by its fishing mortality [18]. *V* is the result of combining productivity (*P*) and susceptibility (*S*) attributes to build a specific score that quantifies the vulnerability associated with a stock. Stocks found to be the most vulnerable to fishing were considered low in productivity and high in susceptibility, while stocks high in productivity and low in susceptibility were deemed the least vulnerable. The Euclidean distance of the weighted average 3–P and S–1 scores from the origin of a biplot of the equation $V = \sqrt{(3 - P)^2 + (S - 1)^2}$ [23] was used to quantify species vulnerability. In this equation, the weighted average *P* scores are shown on the x-axis using a high to low (3→1) scale, and the weighted average *S* scores are plotted on the y-axis using a low to high (1→3) scale. Finally, the vulnerability categories of the bycatches were defined based on the vulnerability scores (*V* < 1.8 = Low, 1.8 ≤ *V* < 2 = Moderate, *V* ≥ 2 = High) proposed by Faruque and Matsuda [33].

### 2.6. Comparison of Species V Score with the Exploitation Rate (E) and Catch Trend

The credibility issues of PSA have been addressed by some authors by comparing their PSA findings with the outcomes of other benchmark methods. PSA findings were previously confirmed, for example, by comparing them to the IUCN Red-List categories, under the premise that species with higher risk ratings belong to these categories (e.g., vulnerable, endangered, critically endangered) [5,26,44]. In addition, PSA results were compared to a proxy of the stock abundance (e.g., catch per unit effort) [45] and stock status based on the ratio of actual fishing mortality to the fishing mortality that yields the maximum sustainable yield [14,44] and historical catch trends (e.g., increasing and decreasing) under the assumption that species with higher risk ranks/values suffer from overfishing or stock depletion [46].

Most of the bycatch species that we selected for our analysis lack national or regional IUCN assessments, although global IUCN risk ranks exist. However, the global IUCN assessment does not always correspond to the national IUCN Red List, and many global IUCN assessments downgraded the species threat rank compared to the national IUCN Red List [33,47]. In the present study, we did not compare our PSA results with the IUCN Red List since the evaluated species did not have national IUCN Red List ranks. Instead, the findings of our PSA (*V* score) were primarily compared with one empirically derived quantitative assessment outcome (i.e., exploitation rate, *E*). This kind of comparison is needed to minimize the uncertainty of PSA outcomes, which will eventually increase the confidence of knowledgeable stakeholders in PSA [5]. This comparison also supports a better understanding of the relative risks confronted by bycatch species due to particular fishing activities. According to Gulland's approximation, the estimated values of the exploitation rate (i.e., the ratio of fishing mortality to total mortality) can be used to assess the overfishing status of a given stock (i.e., when *E* > 0.5) [48]. It was previously suggested that the vulnerability of a stock is directly related to overfishing, and a stock with a *V* score above 1.8 is likely to be associated with an overfishing problem [33,44]. However, it is not always necessarily true that stocks with V > 1.8 are overfished or undergoing overfishing conditions as the *V* score is a relative measure of risk rather than an absolute one and may vary across fisheries [23]. We found a direct relationship between the exploitation rate of the stocks (which quantitatively defines overfishing and underfishing condition) with their corresponding *V* score; therefore, in this analysis, we intuitively assumed that a *V* score of 1.8 is a critical value for the bycatch stocks of Hilsa gillnet fishery of Bangladesh.

The *E* value is typically calculated based on all gear types and thus describes the total fishing mortality of all gear types relative to total mortality. However, the *V* score for a

given species is specific for a particular gear type (Hilsa gillnet in our case). Therefore, some inconsistency between the two outcomes (the *V* score and *E*) is inevitable. We also compared our *V* score with another qualitative indicator, the catch trends of bycatch stock. Species–species catch statistics are unavailable for the majority (20 species out of 21) of the selected bycatches in Bangladesh. Therefore, during the interviews with individual fishers, we asked each of the interviewees about the catch trends of the selected bycatch species. We presumed that experienced fishers' perceptions of catch trends for a given stock would reflect the relative status with greater certainty than other methods. If over 30 Hilsa gillnet fishers (a statistically meaningful majority of 5%) perceived the catch trend for a particular bycatch species to be "decreasing (−1)", "increasing or steady (1)", or "increasing (2)", then we ranked that species as "decreasing", "stable" or "increasing", respectively (Table S1). Any category that did not achieve the consensus of 31 fishers was defined as "not significant (0)". To compare the *V* scores with the catch trends, we assumed that bycatch stocks with "stable", "increasing", or "not significant" trends were subject to underfishing or sustainable fishing, whereas bycatch stocks with "decreasing" catch trends overtime had an overfishing problem.

Finally, we assumed that the higher consistency between the pairs of outcomes (*V* score and *E*; *V* score and catch trends) under two different scoring approaches for PSA would be a useful method in determining the reliable scoring approach for PSA that could be able to minimize the overestimation of species vulnerability.

## 3. Results

The vulnerability scores (*V*) for the 21 bycatch species of the Hilsa gillnet fishery ranged 1.08–2.32 and 1.16–2.38 in ASA and CSA, respectively (Figure 1). The resulting vulnerability scores were used to categorize the bycatch stocks into three distinct vulnerable categories. In CSA, the number of highly vulnerable bycatches increased from two to three compared to ASA (Figure 2). In addition, for two bycatch species (i.e., *Mystus gulio*, BMG; *Pampus chinensis*, CPO), the risk category changed from low to moderate. Ultimately, when CSA was applied to the 21 bycatches, the *V* scores increased by 0–0.20, with a mean value of 0.09 (Figure 1).

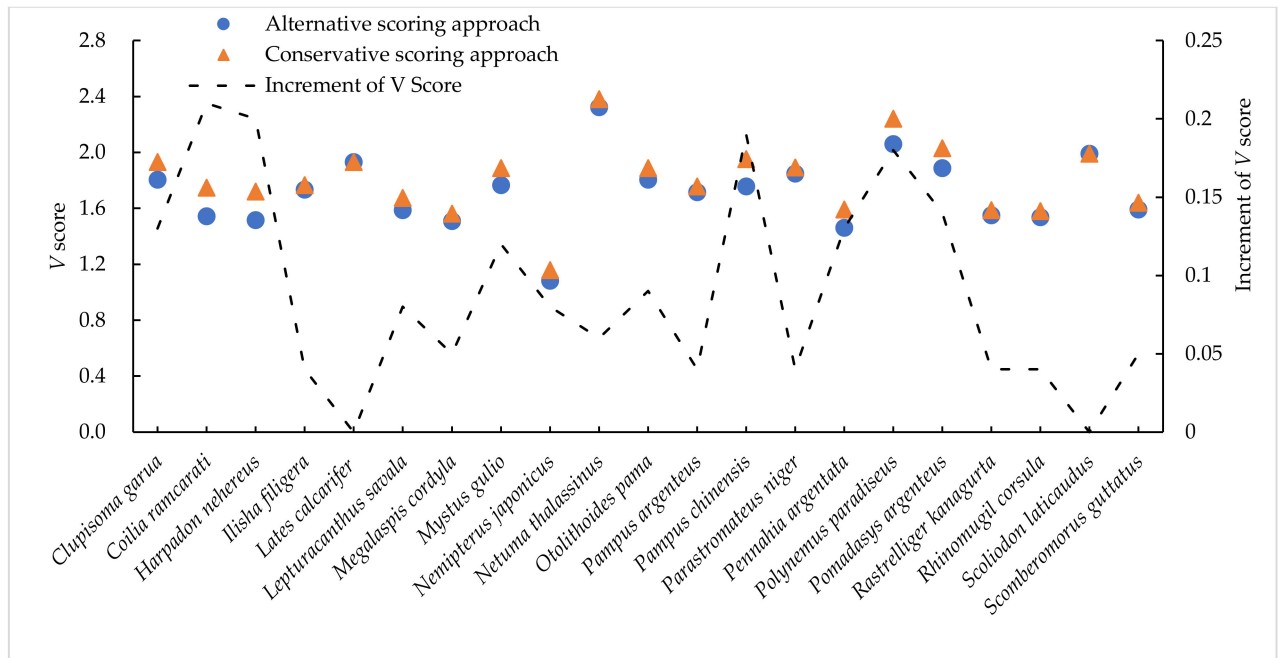

**Figure 1.** Vulnerability (*V*) scores (left y-axis) for the selected bycatch species of the Hilsa gillnet fishery in Bangladesh using conservative and alternative scoring approaches. Values on the right y-axis indicate an increase in the *V* score after applying the conservative scoring approach in PSA.

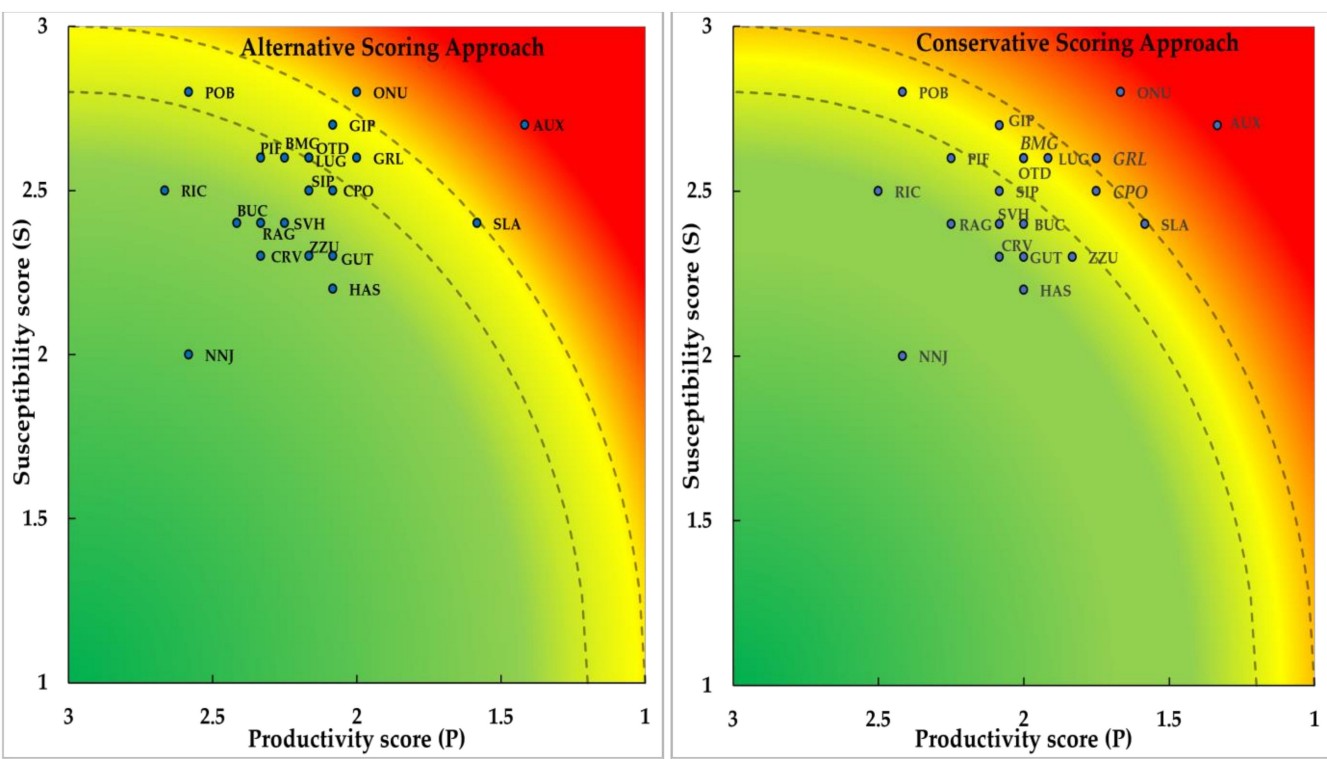

**Figure 2.** Two-dimensional productivity–susceptibility plot for the selected bycatches of the Hilsa gillnet fishery in Bangladesh using an alternative (**a**) and conservative (**b**) scoring approach. The dashed contour lines define the boundaries of the vulnerability categories ($V < 1.8$ = low, $1.8 \leq V < 2$ = moderate, $V \geq 2$ = high). The 3-alpha FAO species identification codes are provided in Table 1. The species codes in italic font (Figure 2b) indicate changes in the species vulnerability ranks between the two scoring approaches.

Figure 3 illustrates a comparison between the *V* score and *E* in ASA and CSA, respectively. Based on the formal stock assessments for the 21 selected bycatch stocks, eight bycatch species (38%) were found to suffer from overfishing, while the remainder 62% (13 in number) suffer from underfishing (when $E > 0.5$, overfishing; $E < 0.5$, underfishing). Following the *V* score and its likely association with the exploitation status, six bycatch species (28.6%) were suggested to have overfishing status, while the remainder of the 15 bycatch species (71.4%) were suggested to have underfishing status when we considered ASA (Figure 3a). Despite being classified as overfished by previous studies based on exploitation rate ($E > 0.5$), our analysis suggests that *Coilia ramcarati* (ZZU), *Harpadon nehereus* (BUC), and *Rastrelliger kanagurta* (RAG) are found to suffer from overfishing. In contrast, our PSA results suggested an overfishing status for *Lates calcarifer* (GIP), but this species was instead given an underfishing ($E < 0.5$) classification based on the exploitation rate. The inconsistency between these two outcomes was 19.0% (four cases out of 21).

On the contrary, when we applied CSA, a total of ten bycatch species (47.6%) were suggested to suffer from overfishing, and the underfishing stock decreased from 15 to 11 (Figure 3b). Our results also suggest that CSA overclassified the fishing status for an additional four species—*Clupisoma garua* (LUG), *Mystus gulio* (BMG), *Otolithoides pama* (OTD), and *Pampus chinensis* (CPO). Eight inconsistent cases (38.1%) were found when we compared the fishing statuses determined by the *E* score and the likely association of *V* with the fishing statuses of the selected bycatch species.

While comparing the *V* scores of the bycatch stocks of our PSA with the catch trends, our analysis indicated species with a *V* score above 1.8 to have a decreasing catch trend in the ASA scenario (Figure 4a). However, species with $V \leq 1.8$ largely presented a stable catch trend. We presumed that species with decreasing catch trends suffer from overfishing

(when *V* > 1.8). Indeed, the consistency levels between the *V*-score-derived fishing status and fishers' perceived catch trends were found to be high in PSA under ASA. However, some inconsistent cases were observed when CSA was applied. For instance, two species with stable catch trends (*Otolithoides pama* and *Pampus chinensis*) and one bycatch with not significantly changed catch trend (*Mystus gulio*) were suggested to be at an overfishing state based on the *V* scores in our PSA (Figure 4b). However, it is reasonable to assume that species with stable catch trends or catch trends without significant changes are sustainably fished or undergoing underfishing but do not suffer from overfishing problems.

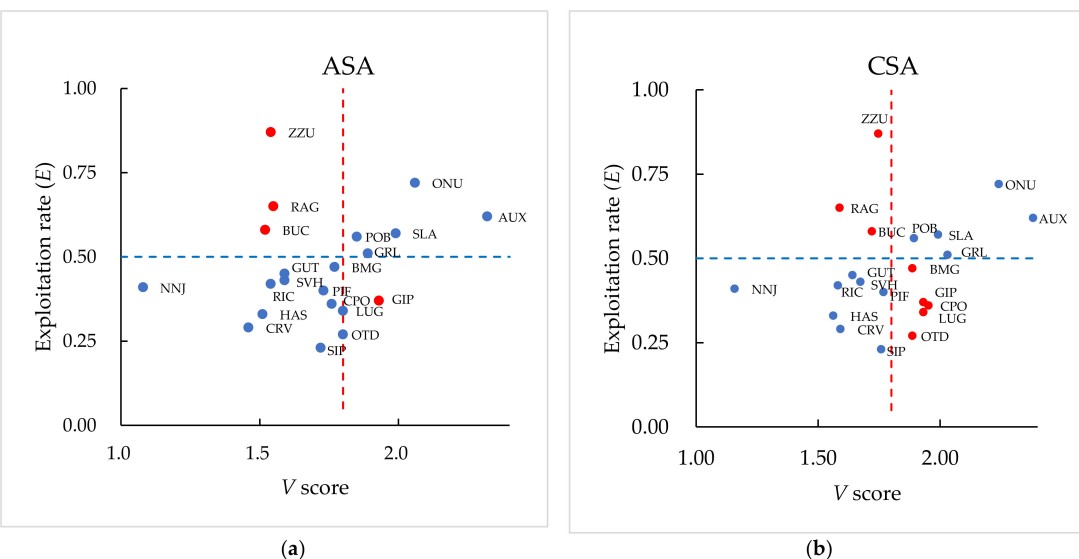

**Figure 3.** A comparison of the exploitation rates (*E*) and vulnerability scores (*V* scores) for the selected bycatches of the Hilsa gillnet fishery in Bangladesh, where the *V* scores were derived from the productivity susceptibility analysis (PSA) under the alternative scoring approach (**a**) and conservative scoring approach (**b**). Blue- and red-colored solid points represent the consistent and inconsistent cases, respectively, between the *E* and *V* scores (when *V* > 1.8 and *E* > 0.5 = overfishing, and *V* ≤ 1.8 and *E* < 0.5 = underfishing). The 3-alpha FAO species identification codes are presented in Table 1.

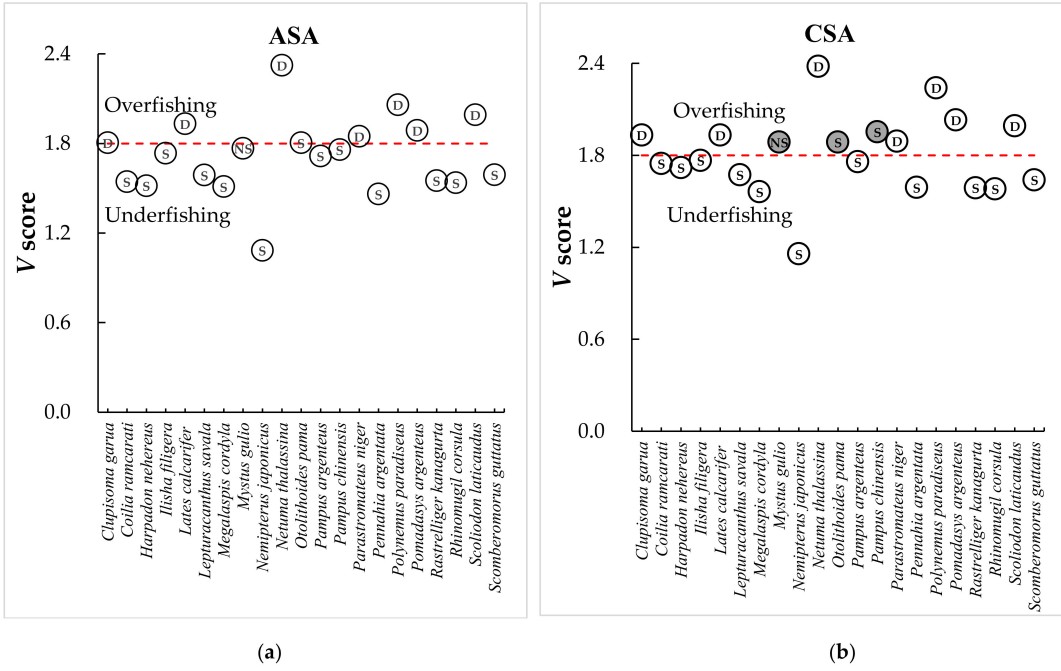

**Figure 4.** A comparison of the catch trends and vulnerability scores (*V* scores) for the selected bycatches of the Hilsa gillnet

fishery in Bangladesh, where the *V* score is derived from productivity susceptibility analysis (PSA) using the alternative scoring approach (**a**) and conservative scoring approach (**b**). The red-colored dashed line represents the demarcation between overfishing and underfishing based on the *V* score (where *V* > 1.8 = overfishing and *V* ≤ 1.8 = underfishing). The points (circle) indicate the *V* score corresponding to each of the assessed stocks, and the letter inside the circle defines the catch trend (D = decreased, NS = not significant, S = stable). Figure 4a indicates that species with a *V* score above 1.8 shows the decreasing catch trend. Figure 4b indicates that the CSA in PSA over-classified the fishing status (i.e., underfishing status to overfishing status) for *Mytus gulio* with a non-significant catch trend and for *Otolithoides pama* and *Pampus chinensis* with stable catch trends (shaded circle).

## 4. Discussion

In response to rising concerns about the impacts of target fisheries on bycatches and associated species, fishery scientists have sought to develop comprehensive risk assessment and management tools for all exploited fishery stocks. PSA is one such tool that can include a large number of exploited stocks in an assessment framework to evaluate the relative risk among species interacting with particular gear types [23]. Despite its extensive usage in fishery sciences for risk assessment, there is no standardized framework for PSA [27]. As a result, risk assessors can tailor the PSA tool in a variety of ways (e.g., for determination of the scoring threshold and treatment of missing information) based on the assessment objectives, fishery characteristics, and data availability [3]. In general, when precise data for the attribute scoring of a species is unavailable (e.g., the $t_{max}$ of a fish determined from otolith or scale methods), PSA may use the imprecise data (e.g., adopting $t_{max}$ data for a species from the same genus or family) and thereby predict species vulnerability. Thus, the uncertainty in PSA outcomes cannot be avoided when low-quality data is used or when the highest score is assigned in the case of unavailable information [5,24]. In the present study, we calculated the vulnerability for the 21 bycatch stocks of the Hilsa gillnet fishery in Bangladesh using two different scoring approaches. Finally, our PSA outcomes were tested against other assessment outcomes to verify which scoring approach is most appropriate in PSA.

Our findings of an increased *V* score and a greater number of moderate and high-risk species under the conservative scoring approach of PSA are consistent with the findings of Osio et al. [44]. Osio et al. applied two scoring approaches—the best guess scoring approach (e.g., using missing information for attributes derived from expert knowledge) and the conservative scoring approach—to study 151 Mediterranean demersal stocks. The authors found that the conservative scoring approach tended to over-classify the risk for many species. The conservative scoring approach generally produced more false positives (i.e., overestimation of risk) than false negatives (i.e., underestimation of risk) [5,49]. We did not find large differences in vulnerability scores between CSA and ASA in our analysis. This result is likely because most of the species-specific information on the life history parameters for the selected bycatch species are available in the existing literature. In the case of data unavailability for 3 (out of 12) particular attributes ($L_{mat}$, $t_{mat}$, $t_{max}$) for some selected bycatches, the assigned scores were changed between the two scoring approaches. However, the resulting vulnerability scores in CSA showed greater inconsistency when compared with other assessment outcomes. Since the contribution of each of the attributes to the overall vulnerability score is minimal [50], it is expected that increasing the number of attributes treated with CSA (for missing information) would result in larger differences in vulnerability scores, especially for data-limited bycatch stocks.

Hobday et al. [24] developed a three-tier hierarchical ecological risk assessment framework (three-tier approach) in which PSA was used to screen out low- and moderate-risk species, with high-risk species suggested for quantitative risk assessment at a higher level. The authors determined that species with over-classified risk ranks (false-positive results) due to the assignment of conservative risk scores would eventually be screened out during higher-level assessments. However, quantitative assessment at a higher level in the authors' proposed framework entails higher data requirements, which are difficult to manage for

a large number of species if many false cases occur in PSA. The data needed for higher-level assessments of a large number of species would take several years to complete and implement [49]. However, Rosenberg et al. [50] argued that the underlying benefit of CSA in PSA is that it provides an incentive to gather more information, and new robust data can only ever decrease the risk score, not raise it.

In PSA using the alternative scoring approach, we found a greater consistency between our V-score-suggested fishing status and the fishing status determined from the exploitation rate, with only a few inconsistent cases. Lucena-Frédou et al. [14] found a similar level of consistency between their PSA findings and exploitation status (fishing mortality relative to fishing mortality that gives the maximum sustainable yield) for finfish caught by pelagic tuna longline fleets in the South Atlantic and Western Indian Oceans. Overall, 75% of stocks with a higher risk score (V = 1.96–2.64) were found to be overfished or subjected to overfishing conditions. Since PSA does not provide any absolute stock estimates, the values of V scores and their likely association with exploitation status can vary between fisheries [23,44]. Osio et al. [44] reported that unsustainable exploitation was mostly observed for Mediterranean demersal stocks with higher *V* scores ($\geq$1.8). Patrick et al. [3] also observed vulnerability ratings greater than 1.8 in 50 American fish populations that had previously been overfished or were presently being overfished.

Fishing mortality varies across gear types, which has a direct influence on the exploitation rate (i.e., fishing mortality relative to total mortality) [48]. Therefore, the inconsistent instances of *Coilia ramcarati* (ZZU), *Harpadon nehereus* (BUC), and *Rastrelliger kanagurta* (RAG) could be explained by a lack of compatibility [51,52]. The majority of the aforementioned species are obtained using other types of fishing gear, such as set bag nets and trawl nets, with different levels of fishing mortality [53]. The disparity for the *Lates calcarifer*, on the other hand, could be explained by the prior study's limited coverage in its sample areas [54]. In contrast, when using PSA with the conservative scoring approach, we observed lower consistency in the *V*-score-suggested fishing status and the absolute fishing status determined by the *E* score from the formal assessment.

The stock status for *Otolithoides pama, Pampus chinensis*, and *Mystus gulio* remained stable and did not significantly change the catch trend, which suggests that these species do not suffer from overfishing problems despite being classified as overfished by their *V* scores under the conservative scoring approach. Although using qualitative catch trend analysis (i.e., the fisher's perceived stock status) to determine the stock status for fishery stocks or populations is not as robust as other quantitative indices such as catch per unit effort [44], the catch trend has been used for many years in fishery science to determine the stock status when there are no quantitative data [55,56].

The PSA results are less precise than those obtained from fully quantitative stock assessments. However, when comprehensive data on stock abundance, catch levels, or other conventional fisheries indicators are lacking, PSA offers a helpful starting point for identifying the relative risk of a species due to fishing, thus prioritizing data collections, future research needs, and management activities. A higher level of agreement between PSA outcomes and the results obtained from other reliable quantitative assessments may increase stakeholders' confidence in PSA's outcomes. The PSA approach performed on 21 bycatch species from the Hilsa gillnet fishery in Bangladesh does not replace the conservative scoring method in PSA but instead provides aid for PSA users to determine which scoring approach is most reliable in PSA. Our PSA outcomes for the two different scoring approaches suggest that the conservative scoring approach could overestimate vulnerability. In contrast, the alternative scoring approach is comparatively more reliable in PSA, which could minimize false estimates of species vulnerability and thus increase the credibility of PSA's application in data-limited situations.

**Supplementary Materials:** The following are available online at https://www.mdpi.com/article/10.3390/fishes6030033/s1, Table S1. Exploitation rate (*E*) of and catch trend of the selected bycatch of Hilsa gillnet fishery of Bangladesh. Species listed in bold text are from inland habitat (river) and the rest of the species are reported from marine habitat, Table S2. Study districts including the survey points details. Values inside the parenthesis indicates the number of participants in each FGD, Table S3. Productivity attributes with values (e.g., $t_{max}$ value), scores (e.g., $t_{max}$ score) and corresponding references used in the productivity susceptibility analysis (PSA) for the selected bycatch of Hilsa gillnet fishery of Bangladesh. Each of the attribute's names in full form is provided in the main text (Table 2). Attributes values have mainly complied from existing literature (normal text). In absence of information for particular attributes (bold italic text), we have assigned scores in both the conservative and alternative scoring methods. Score inside the parentheses is being assigned considering conservative scoring approach, whereas value outside the parentheses is assigned based on corresponding attribute value calculated from empirical relationship equations (described in the main text), Table S4. Susceptibility attributes with scores (e.g., AO score) and corresponding references used in the productivity susceptibility analysis (PSA) for the selected bycatch of Hilsa gillnet fishery of Bangladesh. Each of the attribute's names in full form is provided in the main text (Table 3). Attributes values have mainly complied from existing literature, focus group discussion (FGD) and direct field observation (DO) (normal text). In absence of information for particular attributes (bold italic text), we have assigned scores in both the conservative and alternative scoring methods. Score inside the parentheses is being assigned considering conservative scoring approach, whereas value outside the parentheses is assigned based on expert opinion from key informant interview (KII).

**Author Contributions:** Conceptualization: H.F. and H.M.; Methodology: H.F. and H.M.; Formal analysis: H.F.; Investigation: H.F.; Data curation: H.F.; Writing—original draft preparation: H.F.; Writing—review and editing: H.F. and H.M.; Supervision: H.M.; Funding acquisition: H.F. and H.M. All authors have read and agreed to the published version of the manuscript.

**Funding:** The Graduate School of Environment and Information Sciences, Yokohama National University, Japan, sponsored this study through the Research Promotion Program (grant No 65A0515b). This work was partly supported by JSPS KAKENHI grant (20K06180) to H.M.

**Institutional Review Board Statement:** Not applicable.

**Informed Consent Statement:** Not applicable.

**Data Availability Statement:** Available upon request from the corresponding author of this article.

**Acknowledgments:** We are indebted to all local fisheries officials in the areas we surveyed for their assistance in conducting surveys. We are grateful to all Hilsa fishers for their support and exchanging valuable knowledge about the Hilsa fishery and bycatch by engaging in face-to-face interviews and focus group discussions. We are also thankful to Kendra Karr, Oceans Program, Environmental Defense Fund, USA, who provided us with the Excel edition of the PSA tool and the accompanying guidance notes.

**Conflicts of Interest:** The authors declare no conflict of interest.

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
