# Peer review of "Conservative Scoring Approach in Productivity Susceptibility Analysis Leads to an Overestimation of Vulnerability: A Study from the Hilsa Gillnet Bycatch Stocks of Bangladesh"

_fishes, doi:10.3390/fishes6030033_

Round 1
Reviewer 1 Report
Overview:
The authors present two different ways of treating uncertainty attributes scoring in the Productivity-Susceptibility Analysis of a Bangladeshi gillnet fishery. One approach scores unknown attributes using the most conservative score; the other uses empirical relationships to impute the value. The authors significantly change the common scoring bins, but do a great job of linking the vulnerability score to a measure of overfishing (exploitation) in order to interpret the scores and groundtruth them. The specific treatment of the PSA scoring is thoughtfully done, and I think the comparison of the conservative approach with the empirical approach is a very important demonstration of how that treatment of PSA scoring can matter, and caution should be had when applying or interpreting PSA scores on the CSA approach. Overall, I support the objective of this paper and its execution. Below are some suggestion I hope will improve the overall paper, including the interesting choice of putting the Methods last in the paper, which I do not think works.
I started to edit the grammatical errors (I got through the Introduction), but they accumulated rapidly. An overall readability edit is need to tighten the grammar throughout the paper. This is the main reason I am asking for major revisions.
Major considerations:
- The Methods section is in Section 4 instead of 2. Is that how it should be? Seems an odd choice to put it after the Discussion when we need that section to help interpret the Results.
- Figures 2 and 3 should be combined to show the differences on one plot. You could connect the two methods by a line so readers can track those changes.
- Figure 5 (see also line 179): The caption states that the PSA measures a depleted stock, but that is only true if you assume the fishing rate associated with the PSA has been going on for enough of a time to make the population decline below a target. While this can be an interpretation, the more correct interpretation is whether overfishing (the fishing rate is too high) is currently occurring.
- The Fishlife tool (https://github.com/James-Thorson-NOAA/FishLife) is one of the best ways to access FishBase information and have a taxonomically hierarchical approach to imputing missing life history data. I would highly encourage the authors to look at that tool and compare their inputs in the paper to what FishLife provides and note any differences. They may not be any but I think this would be a more standardized way of dealing with imputing data.
Minor considerations:
- Lines 33-36: I do not agree that you have to have “fully” (not clear what that means, but I think it means “complex’?) quantitative stock assessments to compare indicators to reference points. A well done abundance survey with a baseline value to compare is all you need is some instances. I highly recommend rethinking what this sentence is trying to convey. If it is simply that quantitative methods are typically what is used to management stocks to reference points, that is all that should be said.
Edits/suggestions
- Line 31: “Fisheries management benefits from accurate stock status estimates to apply harvest control rules and meet management objectives. ”
- Line 55: “the ecosystem based fishery management has gained in popularity over the last couple of decades. ” I would avoid pitting it against single species stock assessments as they are not mutually exclusive, nor is it pragmatic to go all in on EBFM.
- Lines 57-59: “both target and non-target species are incorporated in ecosystem management objectives.” Again, single-species assessments can be taken collectively for a more complete management of the system, so not need to suggest that is not the case.
- Lines 59-60: “Fishing that impacts stocks can transform ecosystem functioning by altering predator-prey dynamics.”
- Line 78: “Thousands of stocks…”
- Lines 97: “… the von Bertalanffy growth coefficient (k ; how rapidly a fish reaches its maximum size) tends to be…”
- Line 98: “Stocks…”
- Lines 99: “… tend to have low k”
- Line 111: “The two PSA scoring approaches were… ”
Reviewer 2 Report
While I feel the method being explained in the document is useful and interesting, I have some concerns about the way in which the authors are applying it. They quote the use of PSA analysis at various tuna RFMOs, exercises in which I have been involved, and the use of this method has been greatly debated. The authors acknowledge several times that PSA is a relative risk assessment tool, but then they try to assign the various Vulnerability scores corresponding levels of exploitation. I basically strongly disagree with the results presented between lines 140 - 165. I think this is approach is an erroneous use of the method. By definition exploitation status has to be assessed relative to some level (usually reference points) to assess what is overfished or whether overfishing is occurring. The method of comparing V scores to qualitative or semi-quantitative assessments and then looking for correlations to stock status using vulnerability is problematic. A PSA should be used to assess the relative risk a species has for over-exploitation as is clearly stated by the authors in the paragraph beginning on line 259. It should not in any way confer population status and this should be addressed in the paper. In many cases, management bodies are actually moving away from PSA analysis in data poor situations and rather employing data poor assessment techniques to avoid the exact issues I have raised. PSAs can be useful in prioritising which species should receive management attention. They cannot provide meaningful information on current stock status.
In addition, the order of chapters in the document is confusing. It is strange that the materials and methods are presented after the results and discussion. This makes understand what was done in the study very difficult. I was very confused as to how some of the parameters had been estimated when reading through the results, only to find they had been presented at the end of the document. This, in my opinion, must be corrected to avoid further confusion.
There are some other minor issues with the document, such as conclusions regarding what constitutes a bycatch species as listed between lines 47 and 51 which I disagree with (if a stock is commercialised I no longer believe it is bycatch) but these issues are relatively minor. Also a through editing of the English is required. There are many instances of poor language use which should be corrected before publication.
In general I dont disagree with the methods used in the paper nor the utility of the approach, but I strongly disagree with inferring population status using this approach and would highly recommend this is corrected.
Reviewer 3 Report
General Comments
In general, I found the manuscript original and very well written. Potentially it may represent an important piece of information in support to fisheries management for operators, policymakers and the government agencies in multispecific fisheries characterized by lack of detailed information about non-target fish species. A few suggested amendments/fixes to typos are listed below in my “Specific comments”.
Specific comments
Page 2, line 56: Something appears to be missing in the sentence. Insert “has been” between “fisheries” and “getting”?
Page 7, line 186: Set “Mystus gulio” and “Pampus chinensis” format in Italics.
Page 7, line 199: Replace “its” with “it”.
Page 8, line 203: Replace “determine” with “determined”.
Page 8, line 210: Replace “soring” with “scoring”.
Page 9, line 260: Something seems to be missing in the sentence. Consider in inserting “a tool” between “manager” and “to identify”.
Page 9, line 274: Replace “water” with “waters”.
Page 9, line 274-276: Consider rephrasing as follows: “This transboundary species largely MIGRATES from seawater to the estuarine and riverine ecosystem during ITS spawning time and IT IS LARGELY CAPTURED mostly by gillnets”. But I am not sure I have caught the sense…
Pages 11-12, line 304-306 (Table 2): I guess the correct productivity attributes are tmat/tmax and Lmat/Lmax. Amend in the table, where the ratios are reversed.
Page 12, lines 325-326: Consider replacing the sentence as follows: “Some biological parameters (e.g., maximum age and age at first maturity; maximum size and size at first maturity, etc.) are highly correlated with each other;”.
Page 12, line 355: Delete comma after “Likewise”.
Page 12, line 362: Consider replacing “had” with “having” after “(i.e., ”.
Page 13, line 364: “Hisle”? What is it? Should it be “Hilsa” instead?
Page 13, line 381: Replace “2.6” with “4.6”.
Page 13, line 392: Replace “2.3” with “4.3”.
Page 14, lines 439-441: Consider replacing the sentence as follows: “According to Gulland's approximation, estimated values of the exploitation rate (fishing mortality/total mortality) can be used for assessing the overfishing status (i.e., when E>0.5) of a given stock [33].”.
Page 14, lines 441-444: Consider replacing the sentence as follows: “It was observed that the E associated with overfishing corresponds to high V scores; specifically, the stock having a V score above 1.8 is associated with overfishing problems, whereas a V score below 1.8 is associated with underfishing [24, 43].”.
Round 2
Reviewer 2 Report
I am generally happy with the response to the issues I raised during the first review. Im still not in agreement that vulnerability reflects anything about the fishing status apart from the fact that a high vulnerability would of course potentially open a species to being subject to overfishing with very little fishing pressure or effort. This would of course lead to the relationship noted by the authors, however, the level of fishing pressure that would actually trigger the classification of overfishing is still alarmingly absent and so there is a possibility an incorrect classification to this effect could be made. The reverse is also true, and those species currently listed with lower V scores are being assumed to be not subject to overfishing which would categorically not be true if high fishing pressure is exerted. This is often the case for exploited marine stocks where they are relatively robust but are devastated by high fishing effort. I still think the authors could include a sentence or two to highlight this assumption more explicitly and therefore not over (or under) emphasis the use of the V score to determine fishing pressure. I thoroughly agree this method helps prioritise species of concern and can help focus data collection priorities. As such although I have concerns about this particular conclusion I am happy the publication proceeds with a small addition to clarify the above point. Justification for this can be found in most publications describing semi-quantitative PSA analysis where some level of fishing (either effort or catch) needs to be incorporated into the analysis in addition to assessing the Susceptibility and the Vulnerability.
Author Response
Dear Respected Reviewer
Thank you so much for your kind suggestions and valuable comments on our manuscript. We have included some text to our revised manuscript, particularly in the materials and methods section (Lines 291-300) to justify our assumption, also included some text in the discussion section (Lines: 481-485).
Lines 291-300: It was previously suggested that the vulnerability of a stock is directly related to overfishing, and a stock with a V score above 1.8 is likely to be associated with an overfishing problem [30, 43]. However, it does not always necessarily true that stocks with V>1.8 are overfished or undergoing overfishing conditions as the V score is relative measure of risk rather than absolute one and may vary across fisheries [23]. As we found a direct relationship between the exploitation rate of the stocks (which quantitatively defines overfishing and underfishing condition) with their corresponding V score; therefore, in this analysis, we intuitively assumed that V score of 1.8 is a critical value for the bycatch stocks of Hilsa gillnet fishery of Bangladesh.
Lines 481-485: The PSA results are less precise than those obtained from fully quantitative stock assessments. However, when comprehensive data on stock abundance, catch levels, or other conventional fisheries indicators are lacking, PSA offers a helpful starting point for identifying the relative risk of a species due to fishing thus to prioritize data collections, future research needs and management activities.
Kind regards
Authors